# The added value of devices to pelvic floor muscle training in radical post-prostatectomy stress urinary incontinence: A systematic review with metanalysis

**Giardulli Benedetto**, **Battista Simone**, **Leuzzi Gaia, Job Mirko, Buccarella Ottavia, Testa Marco***

Department of Neurosciences, Rehabilitation, Ophthalmology, Genetics, Maternal and Child Health, University of Genova, Campus of Savona, Savona, Italy

* marco.testa@unige.it

## Abstract

### Purpose

To investigate the role of pelvic floor devices (e.g., biofeedback, electrical stimulation, magnetic stimulation, or their combination) as adjunctive treatments in pelvic floor muscle training (PFMT) in stress urinary incontinence (SUI) after radical prostatectomy.

### Materials and methods

A systematic review with meta-analysis. We searched for randomised controlled trials (RCTs) and prospective non-randomised studies investigating the effectiveness of pelvic floor devices as an adjunctive treatment for SUI symptoms assessed with weight pad-test or standardised questionnaires. To assess the risk of bias (RoB) and overall certainty of evidence, the RoB 2.0 or the ROBINS-I, and the GRADE approach were used.

### Results

Eleven RCTs met our eligibility criteria. One was at a 'low' RoB, one had 'some concerns', while nine were at a 'high' RoB. Two meta-analyses were conducted to analyse the pooled results of six RCTs included. Specifically, two RCTs reported at week 4 with a 1h pad test a mean difference of 0.64 (95% CI = [-13.09, 14.36]), and four RCTs reported at week 12 with a 24h pad test a mean difference of -47.75 (95% CI = [-104.18, 8.69]). The heterogeneity was high in both analyses ($I^2 = 80.0\%$; $I^2 = 80.6\%$). The overall level of certainty was very low.

### Conclusions

In line with our results, we cannot conclude whether pelvic floor devices add any value as adjunctive treatment in the management of SUI after radical prostatectomy. Future studies require more comprehensive and standardised approaches to understand whether these devices are effective.

**Data Availability Statement:** All relevant data are within the paper and its Supporting Information files.

**Funding:** The author(s) received no specific funding for this work.

**Competing interests:** The authors have declared that no competing interests exist.

## Introduction

Stress urinary incontinence (SUI) is a common complication after radical prostatectomy [1]. SUI is caused by both the loss of anterior and posterior anatomical supporting structures and by damages of the pelvic innervations [1, 2], resulting in bladder or urethral sphincter dysfunctions, or both [1]. Generally, 5–35% of men after prostatic surgery report urinary leakage [3], and the 95% of them describe symptoms consistent with SUI [2], even after robotic-assisted surgery [4]. Furthermore, patients also experience a relevant decrease in health-related quality of life (HRQoL) after this surgery [5, 6]. As strongly recommended by the European Association of Urology, pelvic floor muscle training (PFMT) represents the first treatment for post-prostatectomy SUI to speed up the recovery process [7]. Briefly, it consists of repetitive voluntary contractions of muscles involved in continency function, such as bulbocavernosus, striated urethral sphincter and puborectalis [8, 9].

During the early phase of rehabilitation, different types of feedback are adopted to facilitate the pattern of activation of these muscles [10, 11]. Feedback can be provided in different ways. First, it can be successfully provided with the supervision of a healthcare professional who verbally guides the patients [12]. Then, positioning a hand in the perineal area can help to receive manual feedback on the contraction [13]. Finally, feedback is provided using devices capable of delivering visual or audio feedback [14, 15]. These devices are also used to induce a local muscle effect through electrical stimulation to improve SUI symptoms [16]. However, most of these devices require an invasive anal approach that contributes to experience discomfort and to create a barrier to treatment adherence [17, 18]. In addition, their use is still controversial since there is no consensus on their efficacy [19, 20].

Three previous reviews have compared the effect of pelvic floor devices as adjunctive treatments to PFMT in men with urinary incontinence after radical prostatectomy [21–23]. All of them concluded that adopting pelvic floor devices improves urinary incontinence symptoms. However, Zaidan P. and Da Silva E.B. [22] investigated the effectiveness of PFMT with or without electrical stimulation without performing a meta-analysis. Hsu L. et al. [23], in their review with meta-analysis, investigated the beneficial effects of biofeedback-assisted PFMT, but the biofeedback was also intended as verbal feedback, and the interventions were sometimes applied before the prostatectomy. Finally, Sciarra et al. [21] reviewed with meta-analysis to investigate the effects of a biofeedback-guided programme or pelvic floor muscle electric stimulation but did not provide a level of certainty of the reported results. Moreover, none of these reviews focused on SUI symptoms.

In light of the above, we performed a systematic review with meta-analysis to investigate the effect of these devices (e.g., biofeedback, electrical stimulation, magnetic stimulation, or their combination) as an adjunctive treatment in the management of radical post-prostatectomy SUI symptoms.

## Materials and methods

The protocol of this systematic review was registered into the International Prospective Register of Systematic Reviews (PROSPERO; No. CRD42022307289). The reporting of this systematic review followed the Preferred Reporting Items for Systematic Reviews and Meta-Analyses statement (PRISMA) 2020 [24]. To conduct this systematic review, the Cochrane Handbook for Systematic Reviews of Interventions was used [25].

## Objective

The primary aim of this systematic review is to analyse the effect of pelvic floor devices (electrostimulation, magnetic stimulation, and biofeedback] as adjunctive treatments in a PFMT programme in radical post-prostatectomy SUI.

## Eligibility criteria

**Type of publications.**   Only randomised control trials (RCTs) and prospective non-randomised studies were taken into account. No limits on language were set. Case series, single-case studies and systematic reviews were excluded from the analysis. No limitations on the publication date were set. Abstracts and reports from meetings were excluded.

**Population.**   We considered eligible for this systematic review only studies addressing men (age > 18 years) with radical post-prostatectomy SUI. No follow-up, symptoms duration and symptom severity limits were set. We excluded studies where participants had any type of comorbidity that could interfere with the pelvic floor training results (e.g., chronic or acute neurologic diseases, ongoing prostate cancer, surgery neurologic injuries). People with a history of cancer other than recent prostatectomy were excluded as well. Moreover, we also excluded any other UI types, such as urge incontinence.

**Types of intervention.**   Studies that investigated the use of a pelvic floor device (e.g., biofeedback, electrostimulation or magnetic stimulation) as an adjunctive therapy in the management of radical post-prostatectomy SUI were considered eligible. Therefore, studies needed to compare the effectiveness of PFMT with and without the combined use of a device. Any permanent implantable or surgical device was not considered eligible. PFMT was considered as any training involving specifically the contraction of pelvic floor muscles, both supervised and not. No limits on duration or frequency were set. Studies that evaluated a pelvic floor device in isolation were excluded.

**Types of outcomes.**   The primary outcome of this study was the severity of UI symptoms measured either through gold standard objective measures (i.e., pad weight test) or self-reported tools (e.g., international consultation on incontinence questionnaire; ICIQ). No limits on repetitions (e.g., 1h, 4h etc.) were set. The secondary outcome was HRQoL.

**Search strategy.**   As suggested by the Cochrane Handbook for Systematic Reviews of Interventions [25] we chose Pubmed, EMBASE and Cochrane Library-CENTRAL and a specific electronic database based on the research question. Therefore, we also performed our research on PEDro as it is one of the main database for physiotherapy research. We systematically performed the research on these databases up to 12 June 2023.

The search was conducted by three authors (B.G., S.B., and G.L.). The search strategy was a combination of Medical Subjects Headings, Boolean operators (e.g., AND and OR) and the keywords "urinary incontinence", "stress", "prostatectomy", "male", "physical and rehabilitation medicine", "pelvic floor", "exercise", "feedback", "lower urinary tract symptoms", and "incontinence impact questionnaire". The research strings for every database are reported in Supplementary Materials (S1 File).

**Selection process.**   Articles were uploaded onto Rayyan Website after duplicate removal [26]. Afterwards, two researchers (B.G. and G.L.) independently and systematically carried out the starting search applying the inclusion and exclusion criteria to titles and abstracts. When the authors were in disagreements, a third author (S.B.) was consulted to reach a consensus. No author or expert was contacted to get additional studies. When necessary, the full text was read.

**Data collection.**   Two researchers (B.G. and G.L.) independently extracted the following data from each study using standardised Excel templates: authors, year of publication, country, setting, study design, the total number of participants, age, number in each group, type of

intervention and control, the timing of administration of intervention and baseline, post-intervention and follow-up (when available) points estimates, measures of variability of main outcomes and authors key conclusion. Results for both primary and secondary outcomes were extracted. To be able to make a comparison between outcomes and to facilitate the eventual meta-analysis, data were divided based on the times of assessment (e.g., 2 weeks, 4 weeks) and the tests adopted (e.g., 1h pad test, 4h pad test). Authors of studies where data were not completely displayed were contacted. In case of disagreement in the data extraction process, a third author (S.B.) was consulted to gain a consensus.

**Study risk of bias.** The risk of bias and methodological quality of the included studies were independently assessed by two authors (B.G. and G.L.). For randomised controlled trials, we used the Revised Cochrane risk-of-bias tool 2.0 (RoB 2.0), recommended to assess the risk of bias in Cochrane Reviews [27]. This tool allows for assessing on a standard set of items used for the risk of bias appraisal: "bias arising from the randomisation process", "bias due to deviations from intended interventions", "bias due to missing outcome data", "bias in the measurement of the outcome", "bias in the selection of the reported result", and, finally, the risk of bias judgment for each outcome. Instead, we used the Risk of Bias for prospective studies in a non-randomised study (ROBINS-I) [28]. This tool allows for assessing on a standard set of items used for the risk of bias appraisal: "bias due to confounding", "bias in selection of participants into the study", "bias in classification of interventions", "bias due to deviations from intended interventions", "bias due to missing data", "bias in measurement of outcomes" and "bias in selection of the reported result". Both the tools, recommended by the Cochrane Library, allowed the studies to be classified as "low", "some concerns", or "high" risk of bias. In case of disagreement between the reviewers, a consensus was obtained after the consultation of a third one (S.B.).

**Data analysis and synthesis.** Statistical analysis was done via Review Manager 5.3 (RevMan-Copenhagen: The Nordic Cochrane Center, The Cochrane Collaboration, 2014) and Stata 17 (StataCorp). For inter-group comparisons, the mean, the standard deviation, and/or mean differences and the 95% CI were extracted when available or calculated when possible. Medians and interquartile ranges were extracted, when mean, the standard deviation, and/or mean differences were not presented. Continuous data were combined through meta-analysis using a random-effect model when appropriate. As we knew from the literature, that the pad test is the most adopted outcome measure to assess our primary outcome (SUI symptoms), we adopted the 'mean difference' in the meta-analysis as the measures should be comparable. Statistical heterogeneity was assessed using the $I^2$ statistic. The overall certainty of the evidence and strength of the recommendations were evaluated using the Grading of Recommendations Assessment, Development and Evaluation (GRADE) approach [29] through the GRADEpro GDT (https://gradepro.org, accessed date: 12 June 2023), for the primary outcome. The downgrading process was based on five domains: study limitations (e.g., risk of bias), inconsistency (e.g., heterogeneity between study results), indirectness of evidence (generalisability and transferability, e.g., short-term follow-up), imprecision (e.g., small sample size), and reporting bias (e.g., publication bias). A sensitivity analysis was run to evaluate the robustness of our findings. Specifically, we explored the effects of the devices plus PFMT by clustering them based on their type (e.g., biofeedback).

## Results

### Study selection

Database searches initially yielded 4790 articles. After removing the duplicates, they were reduced to 4481. Of these, after the screening selection through titles and abstracts, 23 were

screened for eligibility. Finally, after the full-text screening, only 11 met the inclusion criteria and were considered for our critical review [30–40]. Fig 1 reports the flow diagram that thoroughly reports the study selection process. The studies were published between 1999 and 2023. Of the 11 studies included in the review, all studies were prospective RCTs that had at least two treatments (one with and one without pelvic floor devices). Only one prospective study reached the last screening phase, but it was excluded because there was only one treatment without a control. One study that adopted a subjective outcome measure as primary outcome also reached the final screening phase, however data results were not reported in the manuscript. We contacted the authors for raw data, but they did not get back to us [41]. Therefore, we have excluded this paper as we did not know whether it really answered our research question [41]. Sample size of post-prostatectomy SUI ranged from 13 to 139 across the studies. The pooled population comprised 856 participants. Follow-up during treatment ranged from 1 to 3 months.

## Study characteristic

Table 1 reports the main characteristics of each study. Three studies were conducted in Brazil [30, 32, 38], one in Egypt [40], one in China [39], one in Canada [35], two in South Korea [34, 36], one in Poland [33], one in Greece [37], and one in Germany [31]. As far as the SUI symptoms are concerned, five studies investigated them with a 24h pad test [32, 34–36, 40], four studies with a 1h pad test [37–39, 41], and one with the 20min pad test [31]. Only one study used both the 1h pad test and the 24h pad test [33]. HRQoL was assessed in seven studies out of twelve [30, 32–35, 39, 40], with different scales. Three used the IIQ-7 [32, 33, 40], two used the ICIQ-SF [30, 39], one used both the IIQ-7 and the EORTC QLQ C30 [35], one used the IPSS-QoL [34], and one used I-QOL [36].

## Risk of bias in studies

The risk of bias assessment of RCTs is displayed in Fig 2. Among the included studies, one was at a "low risk", one had "some concerns", and nine were at a "high risk". In general, data were not available for all the participants included in the studies, and there were no analysis methods that correct for bias or sensitivity analysis. Moreover, having a pre-specified analysis protocol was not always possible [31–33, 35, 37–40]. We contacted all authors to receive further information or raw data but only two of them replied without providing any further data [37, 39].

## Results of individuals studies

Available results of individual studies are presented in Tables 2 and 3. Among the studies, there were different treatment arms: seven arms were PFMT not supervised [30–32, 34, 35, 39, 40]; four were PFMT supervised [33, 35, 37, 38]; one was PFMT unclear if supervised [36]; two were no therapy [33, 37]; six were PFMT + Biofeedback (BFB) [32–34, 37–39]; three were PFMT + Electric Stimulation (ES) [31, 35, 40]; two were PFMT + BFB + ES [31, 40]; one was PFMT + Magnetic Therapy (MT) [36]; finally, one was PFP + Pilates [39]. Arms with PFMT supervised and not were considered as control for this review. The treatment methods in terms of frequency and time were different among studies. Sessions lasted from a minimum of 5 minutes to a maximum of 40 minutes.

## Primary outcome–weight pad test

Regarding the weight pad test, four studies found that the intervention group reduced this outcome compared to the control group [32, 33, 39, 40]. On the other hand, one study, as in the

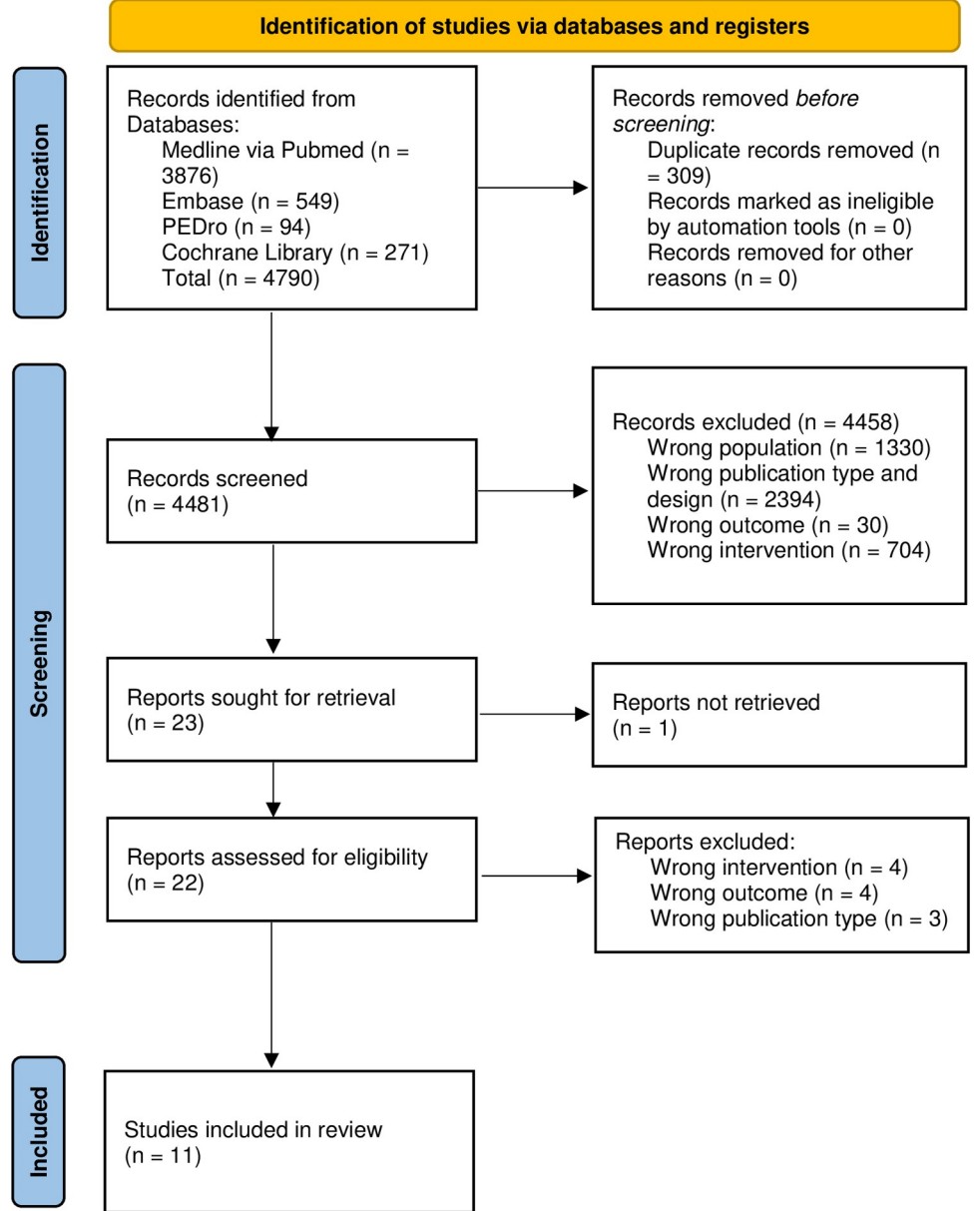

**Fig 1. PRISMA 2020 flow diagram.**

previous studies, found a reduction compared to the control group, but its magnitude was lower [34]. Five studies found no difference between the intervention and control groups [30, 31, 35–37]. Finally, one study [38] reported that the intervention reduced the 1h pad weight compared to control group, but authors did not report the pad weight data in grams, instead they reported a urine severity symptoms classification of participants, based on the amount of pad grams (<2g No UI; 2 to 9.99 g Mild; 20 to 49.9 g Moderate; >50 g Severe), before and after the treatment.

**Table 1. Characteristics of the studies included.**

| Authors | Title | Year | N° | Device used | Intervention | Control | Outcome | Follow-up | Treatment time and n° of sessions |
|---|---|---|---|---|---|---|---|---|---|
| Ahmed M.T., Mohammed A.H. and Amansour A. | Effect of Pelvic Floor Electrical Stimulation and Biofeedback on the Recovery of Urinary Continence after Radical Prostatectomy | 2012 | 80 | A 2-channel electromyographic BFB apparatus was used with one channel for perineal, and the other for abdominal muscles | Group 1: P PFMT + Electric Stimulation Group 2: PFMT + Electric Stimulation + Biofeedback | PFMT not supervised | 24 h pad test IIQ-7 | W0 W6 W12 W24 | 15 min Twice weekly for 12 weeks |
| An D., Wang J., Zhang F., Wu J., Jing H., Gao Y., Cong H., et al. | Effects of Biofeedback Combined With Pilates Training on Postprostatectomy Incontinence | 2021 | 42 | Rectal probe surface electrode | Group B: PFMT + Biofeedback Group C: PFMT + Biofeedback + Pilates | Group A: PFMT not supervised | 1 h pad test ICIQ-SF IEF Oxford Grading Scale | W0 W4 W8 | 40 min Daily for 8 weeks |
| De Santana N.A., De Lima Saintrain M.V., Regadas R.P. et al. | Assessment of Physical Therapy Strategies for Recovery of Urinary Incontinence after Prostatectomy | 2017 | 13 | Anal manometry inflated with 15 ml of air | PFMT + Biofeedback | PFMT supervised | 1 h pad test | W0 W4 W8 | 20 min Once a week for 8 weeks |
| Floratos D.L., Sonke G.S., Rapidou C.A., Alivizatos G.J., et al. | Biofeedback vs Verbal biofeedback as learning tools for pelvic muscle exercises in the early management of urinary incontinence after prostatectomy | 2002 | 42 | A 2-channel electromyographic BFB apparatus was used with one channel for perineal, and the other for abdominal muscles | PFMT + Biofeedback | PFMT supervised | 1 h pad test | M1 M2 M3 M6 | 30 min Three a week for 5 weeks |
| Koo D, Min So S. and Sung Lim J. | Effect of Extracorporeal Magnetic Innervation (ExMI) Pelvic Floor Therapy on Urinary Incontinence after Radical Prostatectomy | 2009 | 32 | Chair with coil-mounted magnets | PFMT + Extracorporeal magnetic innervation therapy | PFMT not specified if supervised | 24 h pad test Number of pads used I-QOL | M0 W1 M1 M2 M3 M6 | 20 min (10 min low frequency + 10 min high frequency) twice a week for 8 weeks + PFMT (10 sets of anal contraction per day) |
| Laurienzo C.E., Magnabosco W.J., Jabur F., Faria E. F., Gameiro M.O., et al. | Pelvic floor muscle training and electrical stimulation as rehabilitation after radical prostatectomy, a randomized controlled trial | 2018 | 123 | Anal electro-stimulator | Group 1: Routine instructions Group 3: PFMT + Electro-stimulation | Group 2: PFMT not supervised | 1 h pad test Perinometers ICIQ-SF IIEF-5 IPSS | M1 M3 M6 | Not specified Twice a week for 7 weeks |
| Moore K.N., Griffiths D. and Hughton A. | Urinary incontinence after radical prostatectomy: a randomized controlled trial comparing pelvic muscle exercises with or without electrical stimulation | 1999 | 58 | Surface anal electrode | Group 2: Intensive PFMT Group 3: PFMT + Electrical stimulation | Group 1: PFMT not supervised | 24 h pad test IIQ-7 EORTC QLQ C30 General Urology Symptom Inventory | W0 W12 W16 W24 | 30 min Twice a week for 12 weeks |
| Oh J.J., Kim J.K., Lee H., Lee S., Jeong S.J., Hong S.K., Lee S.E. and Byun S. | Effect of personalized extracorporeal biofeedback device for pelvic floor muscle training on urinary incontinence after robot-assisted radical prostatectomy: a randomized controlled trial | 2019 | 82 | Portable extracorporeal perineometer | PFMT + Biofeedback | PFMT not supervised | 24 h pad test IPSS IIEF-5 | M0 M1 M2 M3 | 10 min 4 times per day |

(*Continued*)

**Table 1.** (Continued)

| Authors | Title | Year | N° | Device used | Intervention | Control | Outcome | Follow-up | Treatment time and n° of sessions |
|---|---|---|---|---|---|---|---|---|---|
| Rajkowska-labon E., Bakula S., Kucharzexski M and Sliwinski Z. | Efficacy of physiotherapy for urinary incontinence following prostate cancer surgery | 2014 | 81 | Anal probe with electromyography | Group II: no therapy Group IA: PFMT + biofeedback + spinal segmental stabilisation | Group IB: PFMT + spinal segmental stabilisation | 1 h pad test 24 h pad test sEMG muscle tension | Y0 Y1 | 30 min Twice weekly |
| Riberio L.H.S., Prota C., Gomes C.M., de Bessa J., Boldarine M.P., Dall'Oglio M.F., Bruschini H. and Srougi M. | Long-term effect of early postoperative pelvic floor biofeedback on continence in men undergoing radical prostatecomy: a prospective, randomized, controlled trial | 2010 | 73 | Surface anal electrode | PFMT + biofeedback | PFMT not supervised | 24 h pad test ICSI ICST IIQ-7 Oxford scale | M1 M3 M6 M12 | 30 min Once a week until continent or for a maximum of 12 weeks |
| Willie S., Sobottka A., Heidenreich A. and Hofmann R. | Pelvic floor exercises, electrical stimulation and biofeedback after radical prostatectomy: results of a prospective randomized trial | 2003 | 139 | Bioimpulser surface anal electrode | Group 2: PFMT + Electric stimulation Group 3: PFMT + Electric stimulation + biofeedback | Group 1: PFMT not supervised | 20-minute pad test | M0 M3 M12 | 15 min ES 15 min BFB Twice daily for 3 months |

N, number; Int., intervention; Cont., control; W, week; M, month; Y, year; PFMT, pelvic floor muscle training; sEMG, electromyographic signal; ES, electric stimulation; BFB, biofeedback; h, hour; IIQ-7, incontinence impact questionnaire short form; ICIQ-SF, international consultation on incontinence questionnaire short form; IEF, episodes of incontinence; IIEF-5, international index of erectile function; IPSS, international prostatic symptoms score; I-QOL, incontinence quality of life questionnaire; ICIQ, international consultation on incontinence questionnaire; EORTC QLQ C30, european organisation for research and treatment of cancer core quality of life questionnaire; ICSI, incontinence symptoms of the international continence society male short form; ICST, total score of the international continence society male short form questionnaire.

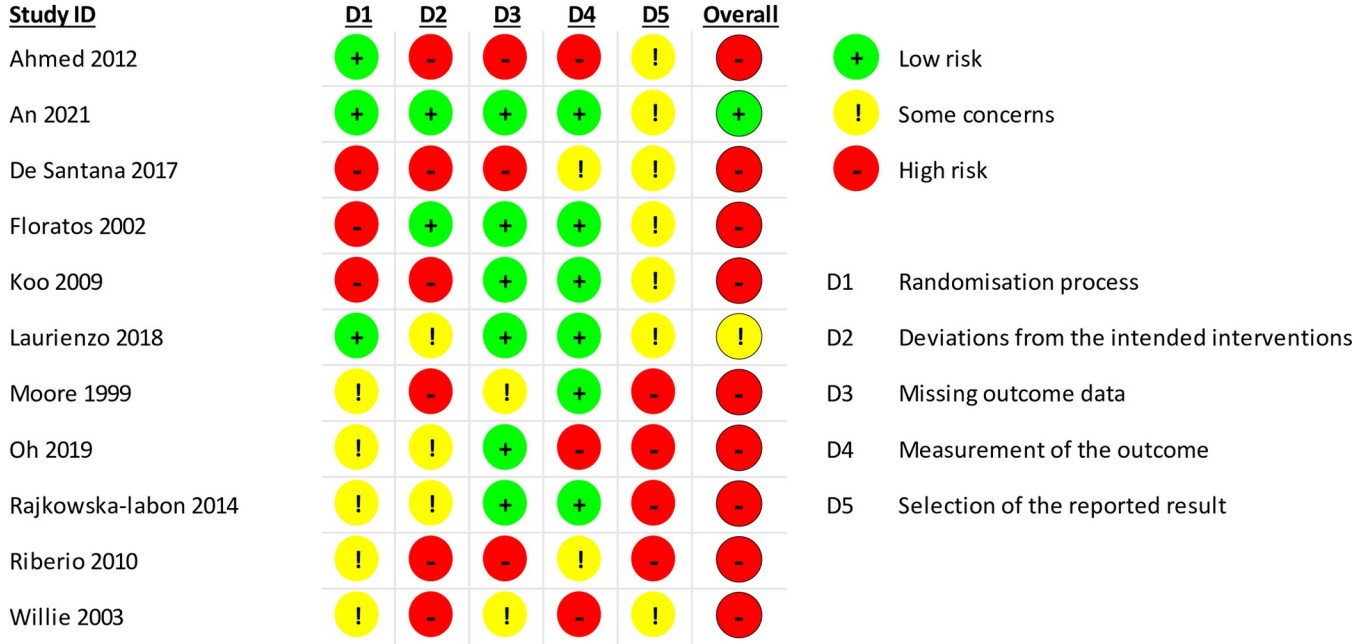

**Fig 2. Risk of bias assessment of the included studies.**

**Table 2. Primary outcome (pad test) of the studies included.**

| Author, year | Outcome measure | Groups | Baseline | T1 | T2 | T3 |
|---|---|---|---|---|---|---|
| **Pad weight test and UI symptoms (Primary outcome)** | | | | | | |
| Ahmed 2012 | 24h pad test | PFMT + ES | 790 ± 399.46 | 383 ± 145.87 | 132 ± 145.87 | 97.8 ± 105.87 |
| | | PFMT + ES + BFB | 785 ± 311.98 | 263 ± 145.87 | 83 ± 145.87 | 36 ± 95.87 |
| | | PFMT NS | 791 ± 380.3 | 533 ± 316.53 | 260 ± 216.53 | 123 ± 116.53 |
| An 2021 | 1h pad test | PFMT + BFB | 58.64 ± 8.72 | 45.93 ± 7.63 | 22.29 ± 4.82 | |
| | | PFMT + Pilates | 56.51 ± 9.46 | 41.43 ± 5.94 | 18.29 ± 2.4 | |
| | | PFMT NS | 57.01 ± 8.46 | 51.46 ± 7.55 | 37.43 ± 7.36 | |
| De Santana 2017 | 1h pad test | PFMT + BFB | No data | | | |
| | | PFMT S | No data | | | |
| Floratos 2002 | 1h pad test | PFMT + BFB | 42 ± 33.33 | 20.52 ± 24.63 | 9.08 ± 12.55 | 7.06 ± 11.7 |
| | | PFMT S | 34 ± 27.44 | 12 ± 11.63 | 5.06 ± 6.49 | 3.7 ± 4.25 |
| Koo 2009 | 24h pad test | PFMT + MT | 436 ± 208 | 147 ± 67 | 9 ± 2.4 | 1 ± 3.3 |
| | | PFMT US | 456 ± 169 | 187 ± 61 | 45 ± 17 | 7 ± 11.7 |
| Laurienzo 2018 | 1h pad test | No intervention | 1 (0–22) | 5 (3–351) | 1 (0–279) | 1 (0–231) |
| | | PFMT + ES | 0.5 (0–36) | 9 (3–241) | 1 (0–183) | 1 (0–18) |
| | | PFMT NS | 1 (0–3) | 7 (3–431) | 2 (0–74) | 1 (0–78) |
| Marchiori 2010 | ICIQ-Male | PFMT + ES | No data | | | |
| | | PFMT NS | No data | | | |
| Moore 1999 | 24h pad test | PFMT S | 565.6 ± 403.3 | 86.9 ± 123 | 73.5 ± 131.4 | 69.9 ± 113.5 |
| | | PFMT + ES | 452.5 ± 385.1 | 155.5 ± 168.1 | 202.2 ± 242.2 | 98.2 ± 131.1 |
| | | PFMT NS | 385.9 ± 256.9 | 103.8 ± 176.3 | 67.3 ± 137.4 | 54.1 ± 103.1 |
| Oh 2019 | 24h pad test | PFMT + BFB | No data | 71 ± 48 | 59.7 ± 83.4 | 38.8 ± 141.2 |
| | | PFMT NS | No data | 120.8 ± 132.7 | 53.1 ± 96.6 | 19.5 ± 57.2 |
| Rajkowska-labon 2014 | 1h pad test | No intervention | No data | | | |
| | | PFMT S | No data | | | |
| | | PFMT + BFB | No data | | | |
| | 24h pad test | No intervention | 61.6 (32.04–12.6) | 12.71 (4.14–17.13) | No data | |
| | | PFMT S | | | | |
| | | PFMT + BFB | No data | | | |
| Riberio 2010 | 24h pad test | PFMT + BFB | 28 (8–82) | 6 (0–24) | 2 (0–12.5) | 0 (0–3) |
| | | PFMT NS | 49 (15–605) | 58 (18–210) | 8 (0–164) | 4 (0–70) |
| Willie 2003 | 20m pad test | PFMT + ES | 35.22 | 75.4 | 80.34 | |
| | | PFMT + ES + BFB | 34.3 | 72.28 | 89.56 | |
| | | PFMT NS | 29.92 | 63.54 | 75.5 | |

UI, urinary incontinence; h, hour; m, minute; T, time; PFMT, pelvic floor muscle training; ES, electric stimulation; BFB, biofeedback; MT, magnetic therapy; NS, not supervised; S, supervised; US, unclear if supervised.

## Secondary outcome–HRQoL

As far as HRQoL, four studies reported that there was no difference between the intervention and control groups [30, 32, 34, 36]. Only two studies reported an increase in HRQoL in the intervention group compared to the control group [39, 40]. Two studies did not provide any raw data concerning this outcome [33, 35]. We have contacted the authors to collected them, but we received no answer.

**Table 3. Secondary outcome (health-related quality of life) of the studies included.**

| Author, year | Outcome measure | Groups | Baseline | T1 | T2 | T3 |
|---|---|---|---|---|---|---|
| | | **HRQoL (Secondary outcome)** | | | | |
| Ahmed 2012 | IIQ-7 | PFMT + ES | 54 ± 26 | 36 ± 25 | 29 ± 28 | 23 ± 24 |
| | | PFMT + ES + BFB | 53 ± 28 | 26 ± 25 | 20 ± 24 | 15 ± 25 |
| | | PFMT NS | 55 ± 31 | 40 ± 23 | 32 ± 26 | 25 ± 26 |
| An 2021 | ICIQ-SF | PFMT + BFB | 16 (16–18) | 12 (10–13) | 8 (7–9) | |
| | | PFMT + Pilates | 17 (16–19) | 10 (11–12) | 6 (5–8) | |
| | | PFMT NS | 17 (16–18) | 15 (13.15) | 12 (11–14) | |
| Koo 2009 | I-QOL | PFMT + MT | 54 ± 13.5 | 79 ± 7.9 | 93 ± 2 | 95 ± 1.2 |
| | | PFMT US | 48 ± 11 | 72 ± 8.7 | 89 ± 4.4 | 93 ± 1.6 |
| Laurienzo 2018 | ICIQ-SF | No intervention | 0 (0–18) | 8 (1–21) | 6 (0–21) | 4 (0–21) |
| | | PFMT + ES | 0 (0–18) | 11 (1–21) | 5.5 (0–20) | 4 (0–18) |
| | | PFMT NS | 0 (0–14) | 11 (1–21) | 6 (0–17) | 3 (0–16) |
| Marchiori 2010 | RAND 36-Item Health Survey | PFMT + ES | No data | | | |
| | | PFMT NS | No data | | | |
| Moore 1999 | IIQ-7 | PFMT S | No data | | | |
| | | PFMT + ES | No data | | | |
| | | PFMT NS | No data | | | |
| | EORTC QLQ C30 | PFMT S | No data | | | |
| | | PFMT + ES | No data | | | |
| | | PFMT NS | No data | | | |
| Oh 2019 | IPSS-QOL | PFMT + BFB | 2.8 ± 1.6 | -1.13 ± 1.65 | -0.9 ± 1.5 | -0.33 ± 1.39 |
| | | PFMT NS | 3.1 ± 1.3 | -0.93 ± 2.02 | -0.57 ± 2.06 | 0.05 ± 2.06 |
| Rajkowska-labon 2014 | IIQ-7 | No intervention | No data | | | |
| | | PFMT S | No data | | | |
| | | PFMT + BFB | No data | | | |
| Riberio 2010 | IIQ-7 | PFMT + BFB | 3* | 2.4 | 0.5 | 0.7 |
| | | PFMT NS | 7.2* | 4 | 2.8 | 1.6 |

HRQoL, health-related quality of life; T, time; PFMT, pelvic floor muscle training; ES, electric stimulation; BFB, biofeedback; MT, magnetic therapy; NS, not supervised; S, supervised; US, unclear if supervised; IIQ-7, incontinence impact questionnaire short form; ICIQ-SF, international consultation on incontinence questionnaire short form; I-QOL, incontinence quality of life questionnaire; IPSS-QOL, international prostatic symptoms score quality of life; EORTC QLQ C30, european organisation for research and treatment of cancer core quality of life questionnaire; *, change in mean.

## Results of synthesis

Based on the availability of the outcome times of assessment and the test used, between the studies, six studies were eligible for meta-analysis [34–37, 39, 40]. Authors were contacted to provide the missing data to possibly extend the meta-analysis. They did not answer or provide the requested data. Two different meta-analyses were conducted based on the availability of the primary outcome data 24h and 1h pad test at 4 and 12 weeks, respectively. Pooled results of the use of a device in addition to PFMT in urinary loss, assessed with weight pad test, reported at week 4 with a 1h pad test a mean difference of 0.64 [95% CI = -13.09, 14.36], and at week 12 with a 24h pad test a mean difference of -47.75 [95% CI = -104.18, 8.69]. The heterogeneity was $I^2 = 80.0\%$; $I^2 = 80.6\%$ respectively. Figs 3 and 4 summarise the results of meta-analyses.

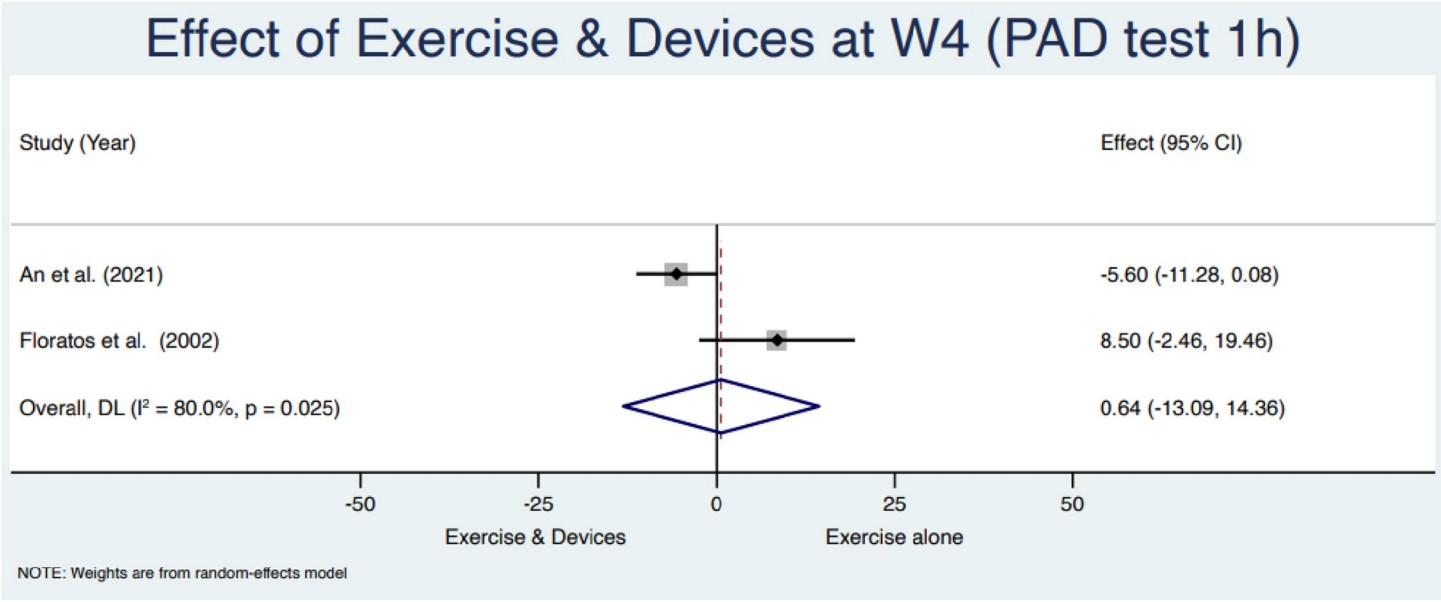

**Fig 3. Meta-analysis for the primary outcome (1h pad test) at week 4.**

## Reporting biases

It was not possible to investigate the bias of publication for the meta-analyses due to the low numbers of studies (<10 studies), as reported in the Cochrane Handbook for Systematic Reviews of Interventions [25].

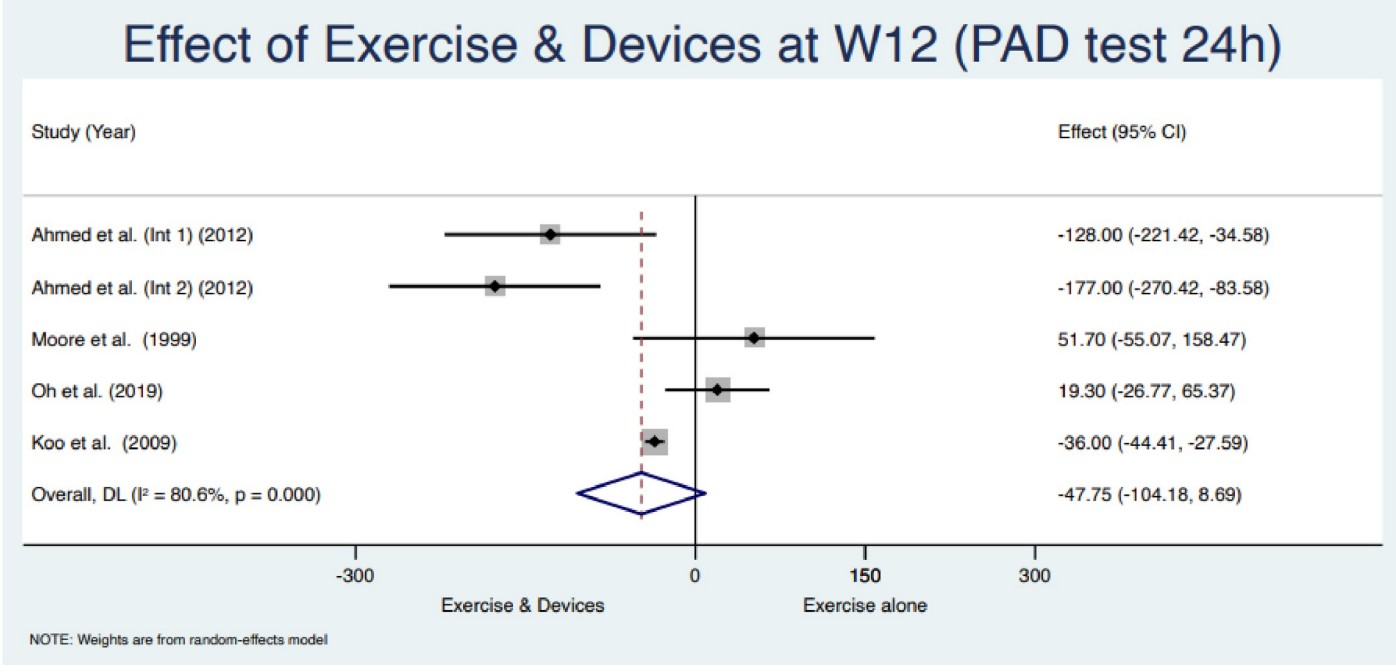

**Fig 4. Meta-analysis for the primary outcome (24h pad test) at week 12.**

## Sensitivity analysis

Based on the used devices, we have divided the studies to run a sensitivity analysis. Among the four studies included in meta-analysis of 24h pad test at week 12, only two adopted the same type of device (electrical stimulation) [35, 40]. The results of the sensitivity analysis were in line with the previous analysis, with a heterogeneity of $I^2$ = 83.8% and a mean difference of -40.08 [95% CI = -216.15, 135.98] (see S2 File for the meta-analysis). We did not run a sensitivity analysis of 1h pad test at week 4 because only two studies were included in the meta-analysis.

## Certainty of evidence

In Table 4 are reported the GRADE assessments. The overall certainty of the evidence was very low.

## Discussion

This systematic review with meta-analysis tested the efficacy of devices as adjunctive treatment to PFMT in the management of radical post-prostatectomy SUI symptoms. Among the eleven studies included in the review, five supported the use of a device in addition to PFMT alone [32–34, 36, 39, 40]. On the other hand, five studies reported no difference between the group with a device and the one with PFMT alone [30, 31, 35–37]. One study did not report raw data on the primary outcome [38]. From the pooled results of the two meta-analyses and the GRADE assessment, we found a high heterogeneity among studies ($I^2$ = 80.0%; $I^2$ = 80.6%] with a level of evidence very uncertain, consistent with the following sensitivity analysis. In line with that, we could not conclude whether the adjunctive use of devices may enhance or not improve SUI symptoms following radical prostatectomy. This finding contrasts with the results of the reviews by Sciarra et al., Silva E.B., and Hsu L. et al. The first review summarised the evidence of the biofeedback and electric stimulation for radical post-prostatectomy UI [21]. In this review, the authors affirmed that the devices in the management of UI following radical prostatectomy improved the incontinence recovery rate within the first 3 months compared with PFMT alone [21]. However, their review was not focused on SUI symptoms and did not provide a level of certainty about the reported results (GRADE approach). The second, instead, investigated the beneficial effects of biofeedback-assisted PFMT, suggesting that biofeedback-assisted PFMT exerts beneficial effects on improving SUI after radical post-prostatectomy [23]. However, they did not focus on SUI symptoms and did not provide a level of certainty about the reported results as well. Conversely, in the review of Zaidan P. and Da Silva E.B. [22] they concluded that electric stimulation associated with PFMT did not show additional benefit. However, they did not perform a meta-analysis. Overall, we can conclude that there is a need of more evidence to understand whether or not these devices are effective as an adjunctive therapy to PFMT.

The huge variability of the results is in line with previous evidence, also about the role of PFMT alone. Cochrane reported that PFMT has conflicting evidence at twelve months post-surgery, concluding that the role of conservative treatment remains uncertain [20]. Conversely, different studies suggest the crucial role of PFMT in recovering or speeding-up SUI symptoms [12, 42–44]. The coexistence of these results may be linked to the different ways in which PFMT is delivered, contributing to the high heterogeneity of PFMT treatment outcomes and study results. Besides the high risk of bias of studies included, three elements discussed hereafter might have contributed to increasing the heterogeneity of our results: supervision, load, and type of PFMT.

**Table 4. GRADE approach assessment.**

| N° of Studies | Study Design | Risk of Bias | Inconsistency | Indirectness | Imprecision | Other Considerations | PFMT + device | PFMT alone | Relative (95% CI) | Absolute (95% CI) | Certainty |
|---|---|---|---|---|---|---|---|---|---|---|---|
| | | Certainty assessment | | | | | N° of patients | | Effect | | |
| colspan | | | | | | | | | | | |

| N° of Studies | Study Design | Risk of Bias | Inconsistency | Indirectness | Imprecision | Other Considerations | PFMT + device | PFMT alone | Relative (95% CI) | Absolute (95% CI) | Certainty |
|---|---|---|---|---|---|---|---|---|---|---|---|
| Urine loss (Follow-Up: twelve weeks; Assessed with: 24h Pad test) | | | | | | | | | | | |
| 4 | RCT | Very serious [a] | Very serious [b] | Not serious | Serious [c] | None | 137 | 139 | // | -47.75 (-104.18, 8.69) | ⊕◯◯◯ Very low+ |
| Urine loss (Follow-Up: four weeks; Assessed with: 1h Pad test) | | | | | | | | | | | |
| 2 | RCT | Serious [d] | Very serious [b] | Not serious | Serious [c] | None | 42 | 28 | // | 0.64 (-13.09, 14.36) | ⊕◯◯◯ Very low+ |

PFMT, pelvic floor muscle training; CI, confidence interval

a downgraded two levels due to different bias

b downgraded two levels due to a considerable heterogeneity of the studies and substantial inconsistency among the

c downgraded one level due to low sample size and contradictory results

d downgraded a level due to different bias from randomisation process, measurement of the outcome and selection of the reported result

† The GRADE approach uses different ⊕ to declare the level of certainty: one ⊕ means very low level of certainty (as in this review), two ⊕ means low, three ⊕ stands for moderate, four ⊕ stands for high.

The first element to consider is whether the PFMT is supervised by a healthcare professional or not, since it has been observed that it may contribute to reaching better outcomes, as shown by Wu M. and colleagues [12]. Among the studies included, there were control groups that received only routine exercises (provided with a pamphlet) compared to interventions with supervised PFMT plus a device. Only four had a control with a supervised PFMT [33, 35, 37, 38]. Instead, one study did not clearly mention if the PFMT control group was supervised [36]. This kind of imbalance between intervention and control groups might have biased the analysis of the results favouring the intervention groups. However, this is just a hypothesis as a few studies reported that routine instructions brought similar levels of improvement compared to supervised PFMT [45, 46].

Secondly, the PFMT load, expressed in time and frequency, might affect the SUI symptoms outcome [47]. Currently, there are no standardised protocols to take as a reference [48]. Therefore, studies adopted different loads. García-Sánchez and colleagues, in their review reported that, independently from the load adopted, all women with SUI improved their symptoms, however they reported that larger effects were reached with a load lasting 6–12 weeks, with >3 sessions/week and a length of session <45 min [49]. The adopted loads by the studies included in our review were all different from each other, and only three of them were in line with the load suggested by García-Sánchez et al. on women [31, 33, 39].

Finally, the intervention itself, based on the provided exercises, may contribute to reaching different outcomes. Hodges P. et al. reported that, to recruit the muscles involved in urinary continence in men better, a focus of anterior pelvic muscles is essential [13]. Therefore, the most relevant command for patients is "shorten the penis" [9]. Moreover, Kruger et al. reported that PFMT requires specifically the muscle recruitment of pelvic floor muscles, and not accessory muscles (e.g., hips, gluteus, abdominals), otherwise the contraction will not be sufficient to bring to an effect [9, 50]. Nevertheless, none of the studies included in our review focused on anterior pelvic muscles. Ten studies reported providing biofeedback or electrical stimulation of anal muscles [30–33, 35–40]. One study adopted control exercises focused on hips, adductors and abdominals [30]. Lastly, one study did not mention exercises on which muscles were focused [34]. Furthermore, given the general trend of results in favour of the

adoption of devices in addition to PFMT, since none of the control groups adopted a sham intervention, it is worth questioning if these results reflected a real efficacy of the devices or a general placebo response [51].

Given the overall controversial results from this review, we wonder whether, in clinical practice, it is helpful to adopt devices with an invasive anal approach in a population who already experience urinary incontinence as a taboo [5, 6]. People with SUI do not consult health professionals for management and treatment due to its negative impact on their privacy and sexuality [52]. The added discomfort experienced from these devices may result in a 'barrier' to a treatment whose efficacy is still controversial. Future studies should adopt sham therapies for control groups to better contain the placebo effect of these devices, but researchers might also give voice to patients to explore the aspects presented above and to understand the perceived usefulness of these treatments. While waiting for future evidence to shed some light on the efficacy of these devices in SUI after prostatectomy, clinicians might opt to choose (or not) these devices based on other factors (e.g., patient preferences).

From the GRADE approach, this review found a very low certainty of evidence. More studies with a robust and accurate design are needed in the scientific literature to shed some light on the real efficacy of devices in the PFMT management of radical post-prostatectomy stress UI. Future studies need to reduce the gaps between the control and intervention arms, to implement a better blinding process, and to define in detail the timing, frequency, and delivery modalities of treatments to ease the comparing process.

Some limits must be reported. First, the high heterogeneity among studies in terms of pad weight ($I^2 > 80\%$), which might be linked to surgical prostatectomy procedures that were not assessed in this study. Secondly, not all studies included reported the characteristics of the population (e.g., age, pad weight) and sometimes the structure of the studies was inconsistent. Studies differed in the reported outcomes and time assessments and in the different interventions delivered, both in modality and load exercise. Finally, we could not assess publication bias due to the low number of studies included in the analysis.

## Conclusions

This review found a very low level of certainty in the evidence and a high level of heterogeneity. Therefore, we cannot conclude whether pelvic floor devices are useful as adjunctive treatment in SUI after radical prostatectomy. Future studies require more comprehensive and standardised approaches to understand whether these devices are effective as adjunctive treatment in this disease.

## Supporting information

**S1 File. Database research strings.**
(DOCX)

**S2 File. Sensitivity analysis.**
(DOCX)

**S1 Checklist. PRISMA 2020 checklist.**
(PDF)

## Author Contributions

**Conceptualization:** Giardulli Benedetto, Battista Simone, Testa Marco.

**Formal analysis:** Giardulli Benedetto, Leuzzi Gaia.

**Investigation:** Giardulli Benedetto, Leuzzi Gaia.

**Methodology:** Battista Simone.

**Project administration:** Giardulli Benedetto, Battista Simone, Testa Marco.

**Writing – original draft:** Giardulli Benedetto, Job Mirko, Buccarella Ottavia.

**Writing – review & editing:** Giardulli Benedetto, Job Mirko, Buccarella Ottavia.

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
