## [Decision Letter · Decision Letter 0]

5 Jun 2023

PONE-D-22-34371The added value of devices to pelvic floor muscle training in post-prostatectomy stress urinary incontinence: a systematic review with metanalysisPLOS ONE

Dear Dr. Testa

Thank you for submitting your manuscript to PLOS ONE. After careful consideration, we feel that it has merit but does not fully meet PLOS ONE’s publication criteria as it currently stands. Therefore, we invite you to submit a revised version of the manuscript that addresses the points raised during the review process.

We look forward to receiving your revised manuscript.

Kind regards,

Ana Katherine Goncalves, Ph.D.

Academic Editor

PLOS ONE

Journal Requirements:

Additional Editor Comments (if provided):

Reviewer 1

Title What type of prostatectomy? Please specify.

Introduction

Specify the prostatectomy you mean. I can guess you are talking about radical prostatectomy. State it clearly

Objective

I think you should state all the once you actually tested. Remove the etc so the work can be reproducible.

Methods

Population: These exclusion criteria were of the studies to make and not you. So, how were you able to determine that the authors did this exclusion if it were not in their exclusion criteria?

Page 5. Primary outcome

Can we have two gold standards in one assessments? Why not stick with the pad test? Pad test is objective while iciq test is subjective. We can only compare studies that used the same outcome measures or validated comparable outcome measures if they are different.

Results

Table 1: Marchiori D., Bertaccini A., Manferrari F, Ferri C. and Martorana G did not state what was used and thus should be excluded. If there is no result on the outcome measure, should it be used for the study. It is an incomplete study and should not be used.

Table 2 De Santana 2017 and Marchiori 2010: If there is no result on the outcome measure, should it be used for the study. It is an incompletes studies and should not be used.

Page 16: Primary outcome – weight pad test

This result is not congruent with the report table where two works did not give data on their primary outcome measure. Santana and Marchiori. Check and explain please.

Discussion

Page 18 2nd paragraph:

State the heterogeneity you noted and why in your estimation you think the studies are faulted to the extent of the information you got. That is what makes it a discussion. Cite reference.

Revise and cite all references.

Reviewer 2

The authors in the meta-analysis sought to evaluate the effectiveness of interventions such as biofeedback and electrical stimulation in controlling stress urinary incontinence after prostatectomy. Eleven clinical trials with more than a thousand patients were included, which confers an important value to the study. However, there is a need for some improvements that I highlight by sections:

Introduction:

It could be improved with one more paragraph, in which the authors should highlight what this systematic review adds to the evidence of others previously published: https://doi.org/10.1590/1980-5918.029.003.AO21 / http://dx. doi.org/10.1016/j.ijnurstu.2016.03.013.

Methodology:

First, the search strategy should be updated with articles published in 2023 and there should not be a limitation of including only articles published in English, as today there are tools that allow good translation into any language.

Sensitivity analysis should be included in the methodology and performed. An example is to perform a sensitivity meta-analysis in the clinical trials of the 24h Pad test subgroup, removing one study at a time and checking for changes in mean difference and heterogeneity.

It is not correct to state that analysis of publication bias cannot be performed due to the small number of trials in the meta-analysis. I suggest that the authors see how to do the analysis with Egger test.

Results

Results after sensitivity analysis will need to be included and whether publication bias exists after applying the Egger test this should also be put in the section.

Discussion

It is well-built, but if new clinical trials published between January 2022 and May 2023 are to be included, it should be reformulated. In addition, a comparison of the results of this meta-analysis with the one I suggested including in the introduction can be carried out (http://dx.doi.org/10.1016/j.ijnurstu.2016.03.013).

Reviewers' comments:

Reviewer's Responses to Questions

**Comments to the Author**

1. Is the manuscript technically sound, and do the data support the conclusions?

Reviewer #1: Partly

Reviewer #2: Partly

2. Has the statistical analysis been performed appropriately and rigorously? 

Reviewer #1: Yes

Reviewer #2: Yes

3. Have the authors made all data underlying the findings in their manuscript fully available?

Reviewer #1: No

Reviewer #2: Yes

4. Is the manuscript presented in an intelligible fashion and written in standard English?

Reviewer #1: Yes

Reviewer #2: Yes

5. Review Comments to the Author

Reviewer #1: Please look at the corrections and comments made in the body of the article. I think with minor corrections as outlined or explanations on the queries raised, the work is good for publication. My recommendations are as stated in the body of the article.

How did you decide on the articles that did not have data on primary outcome measures?

State clearly that you are discussing radical prostatectomy

Reviewer #2: The authors in the meta-analysis sought to evaluate the effectiveness of interventions such as biofeedback and electrical stimulation in controlling stress urinary incontinence after prostatectomy. Eleven clinical trials with more than a thousand patients were included, which confers an important value to the study. However, there is a need for some improvements that I highlight by sections:

Introduction:

It could be improved with one more paragraph, in which the authors should highlight what this systematic review adds to the evidence of others previously published: https://doi.org/10.1590/1980-5918.029.003.AO21 / http://dx. doi.org/10.1016/j.ijnurstu.2016.03.013.

Methodology:

First, the search strategy should be updated with articles published in 2023 and there should not be a limitation of including only articles published in English, as today there are tools that allow good translation into any language.

Sensitivity analysis should be included in the methodology and performed. An example is to perform a sensitivity meta-analysis in the clinical trials of the 24h Pad test subgroup, removing one study at a time and checking for changes in mean difference and heterogeneity.

It is not correct to state that analysis of publication bias cannot be performed due to the small number of trials in the meta-analysis. I suggest that the authors see how to do the analysis with Egger test.

Results

Results after sensitivity analysis will need to be included and whether publication bias exists after applying the Egger test this should also be put in the section.

Discussion

It is well-built, but if new clinical trials published between January 2022 and May 2023 are to be included, it should be reformulated. In addition, a comparison of the results of this meta-analysis with the one I suggested including in the introduction can be carried out (http://dx.doi.org/10.1016/j.ijnurstu.2016.03.013).

6. PLOS authors have the option to publish the peer review history of their article (what does this mean?). If published, this will include your full peer review and any attached files.

Reviewer #1: **Yes: **Timothy Uzoma Mbaeri

Reviewer #2: **Yes: **Ricardo Ney Cobucci

---

## [Author Response · Author response to Decision Letter 0]

6 Jul 2023

RESPONSE LETTER: #PONE-D-22-34371 EMID:553d4bd8f5e8e815

To the kind attention of:

the Editor-in-Chief, Ana Katherine Goncalves;

the Entire review Board of PLOS ONE;

the reviewers: Timothy Uzoma Mbaeri & Ricardo Ney Cobucci.

On behalf of myself and all the authors of the manuscript, we would like to thank you for the effort, professionalism, and time employed in reviewing our study. Based on your constructive and positive comments we have denoted a great improvement in the quality of the work, hoping that it can be considered for publication in your renowned journal.

In the following paragraphs, you will find all the comments highlighted by the reviewers, our answers, and the references to the modification in the text of the revised manuscript. All parts modified or added to the original paper have been highlighted in yellow on the pdf.

Reviewer 1 comments:

“Please look at the corrections and comments made in the body of the article. I think with minor corrections as outlined or explanations on the queries raised, the work is good for publication. My recommendations are as stated in the body of the article.”

Authors’ comments: Thank you for your time and assistance. Based on your feedback we have made some improvements and corrections to the study. In general, we have revised the included studies and decided to implement the studies between January 2022 and June 2023. In this way the review will be more updated. Moreover, based on the suggestion of another reviewer we have run a sensitivity analysis. We hope we have ensured a better quality!

“Title: What type of prostatectomy? Please specify.” 

“Introduction: Specify the prostatectomy you mean. I can guess you are talking about radical prostatectomy. State it clearly”.

Authors’ comments: Thank you for specifying that. We refer to ‘radical prostatectomy’, as you have guessed by reading the manuscript. We have therefore clarified that we referred to ‘radical prostatectomy’.

Authors’ actions: We have added the word ‘radical’ to ‘post-prostatectomy’ or ‘prostatectomy’ all over the manuscript.

“Objective: I think you should state all the once you actually tested. Remove the etc so the work can be reproducible”.

Authors’ comments: We think that you have absolute right here, as it was not clear and reproducible. Therefore, we have removed the ‘etc.’ as you have suggested. 

Authors’ actions: Pag 4, Lines 85-86: the word ‘etc.’ has been removed.

“Methods, Population: These exclusion criteria were of the studies to make and not you. So, how were you able to determine that the authors did this exclusion if it were not in their exclusion criteria?”

Authors’ comments: Thank you for this specification. We could not determine if the authors did this exclusion unless it was declared otherwise. Our intention, in declaring this, was to specify that we would have excluded any study in which these criteria were declared by the authors.

Authors’ actions: We have tried to better explain that these criteria were analysed in the investigating studies and only if these characteristics declared by the authors they were excluded. The paragraph has been amended as follows: Page 4, line 96-101: “We considered eligible for this systematic review only studies addressing men (age > 18 years) with radical post-prostatectomy SUI. No follow-up, symptoms duration and symptom severity limits were set. We excluded studies where participants had any type of comorbidity that could interfere with the pelvic floor training results (e.g., chronic or acute neurologic diseases, ongoing prostate cancer, surgery neurologic injuries). People with a history of cancer other than recent prostatectomy were excluded as well. Moreover, we also excluded any other UI types, such as urge incontinence.”.

“Methods, Primary Outcomes: Can we have two gold standards in one assessments? Why not stick with the pad test? Pad test is objective while iciq test is subjective. We can only compare studies that used the same outcome measures or validated comparable outcome measures if they are different.”

Authors’ comments: We’d like to thank the reviewer for highlight this. Actually, our primary outcome was not the pad test, as this is an outcome measure. Rather, as also reported in the protocol, it was UI symptoms. We have modified the paragraph as reported below. However, we found just one study that used a subjective and validated outcome, but the same study did not report the data. As stated in the paper, we have tried to contact the authors but they did not get back to us. Therefore, the study by Marchiori et al. has been removed, leaving only the studies that adopted the objective pad weight test outcome. 

Authors’ actions: Paragraph ‘Types of outomes’, Pag 5 lines 112-114 has been amended as follows: “The primary outcome of this study was the severity of UI symptoms measured either through gold standard objective measures (i.e., pad weight test) or self-reported tools (e.g., international consultation on incontinence questionnaire; ICIQ). No limits on repetitions (e.g., 1h, 4h etc.) were set.”

The following lines in ‘Results’ section have been added: Page 7, lines 188-192: “One study that adopted a subjective outcome measure as primary outcome also reached the final screening phase, however data results were not reported in the manuscript. We contacted the authors for raw data, but they did not get back to us (41). Therefore, we have excluded this paper as we did not know whether it really answered our research question (41)”

The Marchiori et al. study has been removed from the final included studies.

“Results, Table 1: Marchiori D., Bertaccini A., Manferrari F, Ferri C. and Martorana G did not state what was used and thus should be excluded. If there is no result on the outcome measure, should it be used for the study. It is an incomplete study and should not be used.”

Authors’ comments: We agree with your opinion in this case, since there was no declaration on what was used. For this reason, we have agreed to remove this study from the included studies. 

Authors’ actions: Marchiori et al. study has been removed from the included study.

“Results, Table 2: De Santana 2017 and Marchiori 2010: If there is no result on the outcome measure, should it be used for the study. It is an incompletes studies and should not be used.”

Authors’ comments: As reported above, we have removed Marchiori et al. study. We do not agree with the exclusion of De Santana et al. study, though. De Santana et al. did use the pad test; however, they did not report the pad weight data, but a severity classification based on the pad test, that is, in our opinion a serious reporting bias (‘High’ risk of bias), but not for this reason should be excluded. 

Authors’ actions: Marchiori et al. study has been removed from the included study.

“Results, Page 16: Primary outcome – weight pad test. This result is not congruent with the report table where two works did not give data on their primary outcome measure. Santana and Marchiori. Check and explain please.”

Authors’ comments: thank you for this specification. We have tried to better declare what we have found based on the studies’ reported data. Therefore, we have declared that ‘De Santana’ did report an increase in UI symptoms, but that their reported data were changes in symptoms’ classification based on the amount of pad weights. Marchiori, instead, has been completely removed from the included studies, since authors did not report data (as mentioned above).

Authors’ actions: the ‘Results’ section in Pag 18, lines 2-8, has been amended as follows:

“Regarding the weight pad test, four studies found that the intervention group reduced this outcome compared to the control group (32,33,39,40). On the other hand, one study, as in the previous studies, found a reduction compared to the control group, but its magnitude was lower (34). Five studies found no difference between the intervention and control groups (30,31,35–37). Finally, one study (38) reported that the intervention reduced the 1h pad weight compared to control group, but authors did not report the pad weight data in grams, instead they reported a urine severity symptoms classification of participants, based on the amount of pad grams (<2g No UI; 2 to 9.99 g Mild; 20 to 49.9 g Moderate; >50 g Severe), before and after the treatment.”

“Discussion, Page 18 2nd paragraph: State the heterogeneity you noted and why in your estimation you think the studies are faulted to the extent of the information you got. That is what makes it a discussion.”

Authors’ comments: Thank you for this consideration. We have stated the heterogeneity we noted in the ‘Discussion’ section, and we have tried to better explain what might have affected the heterogeneity of our results (i.e., PFMT supervision, load, type and risk of bias level of studies).

Authors’ actions: Pag 21, Line 8, the I2 Index has been added: I2 = 80.0%; I2= 80.6%.

Pag 21, Lines 26-29 have been amended as follows: “The coexistence of these results may be linked to the different ways in which PFMT is delivered, contributing to the high heterogeneity of PFMT treatment outcomes and study results. Besides the high risk of bias of studies included, three elements discussed hereafter might have contributed to increasing the heterogeneity of our results: supervision, load, and type of PFMT.”.

Pag 22, Lines 58-59 have been amended as follows: “Furthermore, given the general trend of results in favour of the adoption of devices in addition to PFMT, since none of the control groups adopted a sham intervention, it is worth questioning if these results reflected a real efficacy of the devices or a general placebo response (51).”

Pag 23, Lines 67-68, the following sentence has been added: “Future studies should adopt sham therapies for control groups to better contain the placebo effect of these devices, but researchers might also give voice to patients to explore the aspects presented above and to understand the perceived usefulness of these treatments. While waiting for future evidence to shed some light on the efficacy of these devices in SUI after prostatectomy, clinicians might opt to choose (or not) these devices based on other factors (e.g., patient preferences)”.

The following ‘Discussion’ section has been amended as follows: Pag 21, Lines 10-21:

“This finding contrasts with the results of the reviews by Sciarra et al., Silva E.B., and Hsu L. et al. The first review summarised the evidence of the biofeedback and electric stimulation for radical post-prostatectomy UI (21). In this review, the authors affirmed that the devices in the management of UI following radical prostatectomy improved the incontinence recovery rate within the first 3 months compared with PFMT alone (21). However, their review was not focused on SUI symptoms and did not provide a level of certainty about the reported results (GRADE approach). The second, instead, investigated the beneficial effects of biofeedback-assisted PFMT, suggesting that biofeedback-assisted PFMT exerts beneficial effects on improving SUI after radical post-prostatectomy (23). However, they did not focus on SUI symptoms and did not provide a level of certainty about the reported results as well. Conversely, in the review of Zaidan P. and Da Silva E.B. (22) they concluded that electric stimulation associated with PFMT did not show additional benefit. However, they did not perform a meta-analysis. Overall, we can conclude that there is a need of more evidence to understand whether or not these devices are effective as an adjunctive therapy to PFMT.”

“Discussion: Revise and cite all references.”

Authors’ comments: Thank you for this note. We have revised the references and corrected them where necessary. 

Authors’ actions: The following references have been added: 

1. Pag 21, Line 12, 14, the reference n° 21 has been added.

2. Pag 21, Line 19, the references n°22 has been added.

3. Pag 21, Line 17, the references n°23 has been added.

4. Pag 22, Line 59, the reference n° 51 has been added.

 

Reviewer 2 comments:

“The authors in the meta-analysis sought to evaluate the effectiveness of interventions such as biofeedback and electrical stimulation in controlling stress urinary incontinence after prostatectomy. Eleven clinical trials with more than a thousand patients were included, which confers an important value to the study. However, there is a need for some improvements that I highlight by sections:”

Authors’ comments: Thank you for your time and assistance. We followed your suggestions and improved the quality of our study. In particular, we have updated our research until June 2023, and we have improved the introduction and discussions sections. However, we did not feel like to perform a publication bias analysis and due to reasons explained below. We hope that our improvements have increased the quality of our study in a relevant way.

“Introduction: It could be improved with one more paragraph, in which the authors should highlight what this systematic review adds to the evidence of others previously published: https://doi.org/10.1590/1980-5918.029.003.AO21 / http://dx. doi.org/10.1016/j.ijnurstu.2016.03.013.”

Authors’ comments: Thank you for suggesting adding this paragraph. We agree that a paragraph highlighting the adding value of our review would be relevant. Therefore, we have added a new paragraph that discusses what our study adds compared to previous existing studies. 

Authors’ actions: The following paragraph has been added to the ‘Introduction’ section, Pag 3, lines 64-72: “Three previous reviews have compared the effect of pelvic floor devices as adjunctive treatments to PFMT in men with urinary incontinence after radical prostatectomy (21–23). All of them concluded that the adoption of pelvic floor devices is beneficial for improving urinary incontinence symptoms; however, some limits need to be disclosed. Zaidan P. and Da Silva E.B. (22) investigated the effectiveness of PFMT with or without electrical stimulation without performing a meta-analysis. Hsu L. et al. (23), in their review with meta-analysis, investigated the beneficial effects of biofeedback-assisted PFMT, but the biofeedback was intended also as verbal feedback and the interventions were sometimes applied before the prostatectomy. Finally, Sciarra et al. (21) performed a review with meta-analysis to investigate the effects of a biofeedback-guided programme or pelvic floor muscle electric stimulation but did not provide a level of certainty of the reported results. Moreover, none of these reviews focused on SUI symptoms.”

“Methodology: First, the search strategy should be updated with articles published in 2023 and there should not be a limitation of including only articles published in English, as today there are tools that allow good translation into any language.”

Authors’ comments: We have updated our review screening the articles that we rejected previously due to language. Unfortunately, only was study (south Korean) met our inclusion criteria. Moreover, we also run a new research from January 2022 to June 2023. We thank you for that, because with updating our research we have improved the quality of our study!

Authors’ actions: We have included and screened the studies which language was different from English, and we have updated our research until June 2023. The Figure 1 (PRISMA) has been updated. One study has been added into the final included studies. 

“Methodology: Sensitivity analysis should be included in the methodology and performed. An example is to perform a sensitivity meta-analysis in the clinical trials of the 24h Pad test subgroup, removing one study at a time and checking for changes in mean difference and heterogeneity.”

Authors’ comments: Thank you for this consideration. Priorly, we did not declare we would have run a sensitivity analysis on our ‘Prospero Protocol’, still, based on your suggestion, we decided to implement it because we thought it would have been interesting. We could not divide the studies based on the risk of level because they were all at ‘high’ risk, therefore we could not use this criterion for the sensitivity analysis. Moreover, we could not change from a random to fixed effect, since we declared in the protocol that we would have adopted a random one as fixed-effect might not be appropriate based on the evidence on this topic. Finally, we choose not to adopt the ‘leave-one-out’ method for different reasons: potential bias results, lack of justification and limited insight into robustness (https://handbook-5-1.cochrane.org/chapter_9/9_7_sensitivity_analyses.htm). 

For the above-mentioned reasons, we have tried to run a sensitivity analysis dividing the studies based on the device adopted. Fortunately, two studies adopted the electrical stimulation, and therefore comparable. The results were consistent and robust with the overall synthesis of results (Figure 4; 24h Pad test at week 12). 

Authors’ actions: In the ‘methodology’ section, Pag 7 the following lines (175-177) have been added: “A sensitivity analysis was run to evaluate the robustness of our findings. Specifically, we explored the effects of the devices plus PFMT by clustering them based on their type (e.g., biofeedback).”

The sub-paragraph ‘Sensitivity analysis’ has been added to the ‘Results’ section, Pag 18-19, Lines 30-36: “Based on the used devices, we have divided the studies to run a sensitivity analysis. Among the four studies included in meta-analysis of 24h pad test at week 12, only two adopted the same type of device (electrical stimulation) (35,40). The results of the sensitivity analysis were in line with the previous analysis, with a heterogeneity of I2 = 83.8% and a mean difference of -40.08 [95% CI= -216.15, 135.98] (see Supplementary material 2 for the meta-analysis). We did not run a sensitivity analysis of 1h pad test at week 4 because only two studies were included in the meta-analysis.”

The file ‘Supplementary material 2’ containing the results of the sensitivity analysis of the ‘24h Pad test at week 12’, subdivided by device adopted, has been added to the manuscript files. 

“Methodology: It is not correct to state that analysis of publication bias cannot be performed due to the small number of trials in the meta-analysis. I suggest that the authors see how to do the analysis with Egger test.”

Authors’ comments: We would like to thank the reviewer for highlighting this. Cochrane Library states that funnel plot asymmetry should be used only when there are at least 10 studies included in the meta-analysis, otherwise the test’s power will be too low. We did include 11 studies in the systematic review, but not in the meta-analyses, which were fewer than 10. In addition, the Cochrane reports that the Egger’s test is potentially misleading for continuous outcomes, which were our cases. Please, have a look at the following references.

References: https://training.cochrane.org/resource/identifying-publication-bias-meta-analyses-continuous-outcomes

https://training.cochrane.org/handbook/current/chapter-13

https://www.sciencedirect.com/science/article/pii/S0895435699002048?via%3Dihub

Authors’ actions: No actions were taken.

“Results: results after sensitivity analysis will need to be included and whether publication bias exists after applying the Egger test this should also be put in the section.”

Authors’ comments: Thank you for specifying this. Based on the adopted methodology for the sensitivity analysis, explained above, we have conducted and commented the results of the analysis. Unfortunately, we could not match many studies based on the device adopted, therefore only two included studies were eligible for the analysis. The results, anyway, were consistent with the synthesis of results. 

Authors’ actions: The sub-paragraph ‘Sensitivity analysis’ has been added to the ‘Results’ section, Pag 18-19, Lines 30-36: “Based on the used devices, we have divided the studies to run a sensitivity analysis. Among the four studies included in meta-analysis of 24h pad test at week 12, only two adopted the same type of device (electrical stimulation) (35,40). The results of the sensitivity analysis were in line with the previous analysis, with a heterogeneity of I2 = 83.8% and a mean difference of -40.08 [95% CI= -216.15, 135.98] (see Supplementary material 2 for the meta-analysis). We did not run a sensitivity analysis of 1h pad test at week 4 because only two studies were included in the meta-analysis.”

The following sentence has been added to the ‘discussion’ section, Pag 21, lines 6-8: “From the pooled results of the two meta-analyses and the GRADE assessment, we found a high heterogeneity among studies (I2 = 80.0%; I2= 80.6%) with a level of evidence very uncertain, consistent with the following sensitivity analysis.”

“Discussion: It is well-built, but if new clinical trials published between January 2022 and May 2023 are to be included, it should be reformulated. In addition, a comparison of the results of this meta-analysis with the one I suggested including in the introduction can be carried out (http://dx.doi.org/10.1016/j.ijnurstu.2016.03.013).

Authors’ comments: Thank you for your suggestions. We have updated the discussion section adjusting the paragraph that compares our review results to previous reviews including the two reviews you have suggested. Unfortunately, there were no eligible studies between January 2022 and June 2023, therefore we found no need to reformulate this section.

Authors’ actions: The following ‘Discussion’ section has been amended as follows: Pag 21, Lines 10-21:

“This finding contrasts with the results of the reviews by Sciarra et al., Silva E.B., and Hsu L. et al. The first review summarised the evidence of the biofeedback and electric stimulation for radical post-prostatectomy UI (21). In this review, the authors affirmed that the devices in the management of UI following radical prostatectomy improved the incontinence recovery rate within the first 3 months compared with PFMT alone (21). However, their review was not focused on SUI symptoms and did not provide a level of certainty about the reported results (GRADE approach). The second, instead, investigated the beneficial effects of biofeedback-assisted PFMT, suggesting that biofeedback-assisted PFMT exerts beneficial effects on improving SUI after radical post-prostatectomy (23). However, they did not focus on SUI symptoms and did not provide a level of certainty about the reported results as well. Conversely, in the review of Zaidan P. and Da Silva E.B. (22) they concluded that electric stimulation associated with PFMT did not show additional benefit. However, they did not perform a meta-analysis. Overall, we can conclude that there is a need of more evidence to understand whether or not these devices are effective as an adjunctive therapy to PFMT.”

---

## [Decision Letter · Decision Letter 1]

24 Jul 2023

The added value of devices to pelvic floor muscle training in radical post-prostatectomy stress urinary incontinence: a systematic review with metanalysis

PONE-D-22-34371R1

Dear Dr. Marco Testa

**Article type:** Research Article 

We’re pleased to inform you that your manuscript has been judged scientifically suitable for publication and will be formally accepted for publication once it meets all outstanding technical requirements.

Kind regards,

Ana Katherine Gonçalves, Ph.D.

Academic Editor

PLOS ONE

Additional Editor Comments (optional):

The authors followed of the reviewers' recommendations and the manuscript is appropriate for publication.

Reviewers' comments:

Reviewer's Responses to Questions

**Comments to the Author**

1. If the authors have adequately addressed your comments raised in a previous round of review and you feel that this manuscript is now acceptable for publication, you may indicate that here to bypass the “Comments to the Author” section, enter your conflict of interest statement in the “Confidential to Editor” section, and submit your "Accept" recommendation.

Reviewer #2: All comments have been addressed

Reviewer #3: All comments have been addressed

2. Is the manuscript technically sound, and do the data support the conclusions?

Reviewer #2: Yes

Reviewer #3: Yes

3. Has the statistical analysis been performed appropriately and rigorously? 

Reviewer #2: Yes

Reviewer #3: Yes

4. Have the authors made all data underlying the findings in their manuscript fully available?

Reviewer #2: Yes

Reviewer #3: Yes

5. Is the manuscript presented in an intelligible fashion and written in standard English?

Reviewer #2: Yes

Reviewer #3: Yes

6. Review Comments to the Author

Reviewer #2: The authors met most of the reviewers' recommendations and the manuscript is in better shape for publication. Congratulations.

Reviewer #3: The authors followed of the reviewers' recommendations and the manuscript is appropriate for publication.

7. PLOS authors have the option to publish the peer review history of their article (what does this mean?). If published, this will include your full peer review and any attached files.

Reviewer #2: **Yes: **Ricardo Ney Cobucci

Reviewer #3: No

---

## [Editor Report · Acceptance letter]

26 Jul 2023

PONE-D-22-34371R1 

The added value of devices to pelvic floor muscle training in radical post-prostatectomy stress urinary incontinence: a systematic review with metanalysis 

Dear Dr. Marco:

I'm pleased to inform you that your manuscript has been deemed suitable for publication in PLOS ONE. Congratulations! Your manuscript is now with our production department. 

Kind regards, 

on behalf of

Professor Ana Katherine Gonçalves 

Academic Editor

PLOS ONE